# Breastfeeding: A Cornerstone of Healthy Sustainable Diets

**Marian E. Davidove * and Joseph W. Dorsey** 

Patel College of Global Sustainability, University of South Florida, Tampa, FL 33620, USA
* Correspondence: Mdavidove@mail.usf.edu; Tel.: +1-813-789-8356-1540

**Abstract:** On a global scale, the world faces impending food scarcity due to rapid population growth and the deleterious impact of climate breakdown on food production. In the absence of radical change, the most vulnerable and detrimentally affected could be the 2 billion additional inhabitants expected in the developing nations between now and 2050. A root cause of this future scenario is decreasing breastfeeding rates. As the Sustainable Development Goal of Zero Poverty brings the poor in these regions into the middle-classes, there will be an assimilation of Western dietary patterns such as formula feeding and increased intake of livestock and their by-products. Infant formula, the most common alternative to breastmilk, consequently emerges as a formidable driver in the compromise of global food, energy, and water systems. The enormous, intensive water consumption, extensive use of materials for packaging, high-demand use of energy resources in manufacturing, greenhouse gas (GHG) emissions from food miles transportation, and widespread generation of household waste make infant formula production a major environmental concern and a leading contributor to global heating. Exacerbated by population growth, using infant formula to replace breastfeeding irreparably harms societies, economies, and the environment around the world. There is an urgency in addressing the global sustainability impact of using infant formula to replace breastfeeding. It is the purpose of this commentary to demonstrate the social, economic, and environmental costs of using infant formula to replace breastfeeding and provide sufficient evidence to promote breastfeeding as the universal foundation of healthy sustainable diets.

**Keywords:** Consumption and diets; eating patterns; sustainability; food security; dietary intake; breastfeeding; climate-friendly diets; GHGE

---

## 1. Introduction

On a global scale, the world faces impending food scarcity due to rapid population growth with predictions exceeding 9 billion inhabitants by the mid-21st century. A root cause of this growth trend is decreasing breastfeeding rates [1,2]. Studies have shown that using infant formula to replace breastfeeding increases fertility rates and could boost the number of expected infants born by as much as 50% in countries with very high breastfeeding rates [1]. In the absence of radical change, the most vulnerable and detrimentally affected could be the 2 billion additional inhabitants expected in the developing nations between now and 2050 [3]. As the Sustainable Development Goal of Zero Poverty brings the poor in these regions into the middle-classes, there will be an assimilation of Western dietary patterns such as formula feeding and increased intake of livestock and their by-products. Infant formula, the most common alternative to breastmilk, consequently emerges as a formidable driver in the compromise of global food, energy, and water systems. Failing to improve breastfeeding rates and an unchecked formula industry will divert food production, energy, and freshwater to livestock for infant formula production, making it impossible to mitigate the impending food gap, population growth, and climate change. "The world faces a 69% gap between crop calories produced in 2006

and those most likely required in 2050" with the most vulnerable region predicted to be sub-Saharan Africa [3].

The Rio+20 Conference on Sustainable Development Goals (SDGs) [4] in 2012 developed a framework to guide the implementation of sustainable development. There are seventeen SDGs that build upon the Millennium Development Goals and comprise the basis for the 2030 Agenda for Sustainable Development. The SDGs do not distinguish breastfeeding as an individual goal, which is alarming given the convincing evidence that greenhouse gas (GHG) emissions from formula production impact climate change, and considering that the formula market this year will be double what it was in 2007 [4]. When researching the 2012 conference and its goals proposed for 2030, there was no match for the words "breastfeeding goals." However, there appear to be breastfeeding goals embedded or implied in the seventeen SDGs. It is the vision of this author to propose a separate Sustainable Development Goal on breastfeeding providing a necessary framework on corrective action, within the next decade. It is also the purpose of this commentary to provide sufficient evidence to promote this behavior as a universal element of healthy sustainable diets. After a brief history of infant feeding practices, the social, economic, and environmental costs of using infant formula to replace breastfeeding will be described from a global perspective first and an American perspective second.

## 2. A Historical Perspective on Infant Feeding

Before 10,000 B.C., safe and complete alternate infant feeding methods were not available. If an infant could not breastfeed, there were few options and prognosis was poor. If a mother could not or did not want to breastfeed, she would have to find a wet-nurse to nourish her baby. The advent of domestication of grains and animals allowed infants and children to derive adequate nourishment from either gruel, another mammal's milk, or a combination of both. With limited knowledge of infant nutritional requirements and sanitation, these alternate sources were either inadequate or contaminated by dirty water or food-borne pathogens. In prehistoric times, the health outcome was poor in infancy. However, gruel was an acceptable breastmilk substitute for older children, allowing women to wean two years earlier. Weaning a toddler at two instead of four years of age increased fertility to a degree that the world's "virtually static population began slowly to grow, from an estimated five million worldwide in 10,000 B.C. to approximately twenty million by 5000 B.C" [5].

Until the last one hundred years, the normal mode of infant feeding was breastfeeding and all of human beings' evolutionary leaps and advances were partially enabled by this natural, complex, and sustainable process. The Age of Enlightenment brought with it the first patented and marketed alternative to breastmilk. German chemist and professor at the University of Giessen, Justus Von Liebig, developed a liquid and powdered cow's milk enriched with wheat or malt flour and potassium bicarbonate [2]. This idea took off with many similar products being offered on the market in the late 1800s. Brands such as Nestle's Food® and Eagle Brand Condensed Milk® offered Europeans and Americans an alternative to breastfeeding. With the addition of vitamin fortification and sterilization techniques, infant formula became accepted as a safe and popular feeding method receiving approval from the American Medical Association in 1929 [2].

## 3. Global Social Issues

The choices and behaviors of a mother–infant dyad drive population growth trends and establish widespread dietary patterns. In most Western families, the mother is the person who makes decisions regarding food and meals. It is this woman who goes to the store more often in a division of gender roles that remains conventional. She decides where to shop and how frequently. She decides how to feed her infant and children, choosing between modes of breastfeeding and formula feeding from the very beginning. It is important to recognize the urgency of affecting this key decision-maker in the home.

Throughout the past century, mothers became consumers, their choices informed by "scientific evidence" and massive infant formula marketing campaigns. Western societal opinions metamorphosed

dramatically to view breastfeeding as primitive and inferior, and sometimes a socially unacceptable behavior. Breastfeeding rates subsequently plummeted worldwide. Could the evolution and permeation of formula feeding during this time frame be partly responsible for the population explosion known as the baby-boom? The mid-century step-increase in population growth overlapped with a cultural movement away from breastfeeding. Hence, post-World War II population momentum ran parallel to a decline in breastfeeding initiation and exclusive breastfeeding of infants under 6 months of age, compounding a vast array of public health concerns.

## 4. U.S. Social Issues

Mid-century feminist social movements pushing for equality in the workplace and greater educational opportunities also directly and indirectly undermined breastfeeding. By the end of the 20th century, in the United States, only 42% of women were exclusively breastfeeding [2]. Ironically, research conducted by Best Start© [6] in 1990 listed the major social barriers to breastfeeding in the southeastern United States as lack of confidence, embarrassment, fear of loss of freedom, concerns about a "too strict" diet, and the influence of family and friends. Therefore, women actually lost power and self-worth as a result of normalizing formula feeding and the departure from what is natural.

Today, women in lower economic brackets have less access to healthy nutritious foods and fewer resources available to manage household budget. The food budget is often the only line item that can be amended after paying fixed expenses such as housing, electricity, water, and transportation. To compound the issue, dependence on formula introduces a greater possibility of food insecurity by diverting food dollars away from fresh, whole foods. Along with greater chance of food insecurity, lower-income mothers and infants who formula feed suffer greater associated health implications [7]. Mothers who do not breastfeed have an increased risk of premenopausal breast cancer, ovarian cancer, retained gestational weight, type 2 diabetes, and metabolic syndrome. Infants who do not breastfeed have an increased risk of pneumonia, gastroenteritis, otitis media, childhood obesity, type 1 and type 2 diabetes, leukemia, sudden infant death syndrome (SIDS), and necrotizing enterocolitis (in premature infants not receiving breastmilk) [7]. As low income, African American and Hispanic groups are least likely to breastfeed, this issue can be seen as a function of social injustice and health inequity in the United States. Using infant formula to replace breastfeeding has negatively affected society by contributing to unsustainable population growth, destabilization of the mother–infant dyad, food insecurity, and negative health outcomes.

## 5. Global Economic Issues

Globally, not breastfeeding is a costly diversion of funds. According to a recent article exploring the cost of not breastfeeding in Southeast Asia, an additional contributor to economic losses is a decrease in cognitive abilities. Estimates arrive at a significant 0.5% loss of gross national income in Thailand [8].

An economic gender equality issue arises as women pay a price for breastfeeding when they must take leave of absence from work, make sacrifices at work, and choose different career paths to enable their ability to breastfeed successfully. In Scandinavia and other countries with longer maternity/parental leave, the incidence of exclusive breastfeeding and breastfeeding duration increases, underlining the importance of institutionalized financial support [9]. Breastfeeding is undermined quickly and effectively by poor hospital practices and marketing strategies such as providing free formula disguised as gifts. Moreover, infants quickly adapt to the feeding strategy that provides the most energy with the least effort, the "optimal foraging behavior" [5]. Meanwhile, breastfeeding provides food sovereignty and puts individuals in control of their own nutritional well-being, not governments, manufacturers, and marketing institutions. This brief analysis is an example of the deleterious effects on society by a novel approach to infant feeding.

## 6. U.S. Economic Issues

According to the United States Department of Agriculture (USDA), "a minimum of $3.6 billion would be saved if breastfeeding were increased from current levels to those recommended by the U.S. Surgeon General" [10]. Both direct and indirect savings would stem from decreased disease, health, and household expenses. Serving nearly 8 million people, Women Infants and Children (WIC), a special supplemental food program of the USDA, provides infant formula at no cost to low-income participants. In 2016, infant formula company rebates of $1.7 billion reduced total WIC food costs of $5.6 billion by nearly a third [11]. "The WIC program spends more on formula than any other food—$927 million in the fiscal year 2010. The cost of formula contributes to over a third of the overall WIC budget" [11]. In a recent USDA report, "approximately 31.7% of WIC infants were reported as breastfed nationally (13.2% were fully breastfed and 18.5% were partially breastfed)" [12].

## 7. Environmental Issues

Compared to the social and economic impacts of using infant formula to replace breastfeeding, environmental harm is the most difficult to mitigate and therefore the most pressing concern. It is also important to treat the environment as a common resource. Detrimental or mitigating factors cannot be isolated, as impacts in one country affect the health and wellbeing of those in other countries. Therefore, the final topic of this commentary will be addressed as a unified concern.

Breastmilk is the ultimate sustainable food source with a near-zero ecological and carbon footprint. Only 500 additional calories and approximately one liter of additional drinking water per day are needed for lactation. However, the environmental impact of using infant formula to replace breastfeeding is a leading driver in the compromise of the earth's interlinked water, energy, and food systems. As breastfeeding is essentially a provisioning eco-service that is difficult to monetize, an alternate method to demonstrate the value of breastfeeding is an estimation of the environmental cost or negative externalities of formula feeding.

A carbon footprint and water footprint are insightful tools for quantifying the toll of infant formula production on the environment. An analysis of the common components of infant formula as listed by the Codex Alimentarius, a complete registry of ingredients for standard food products, is one way to measure the energy and water needed to synthesize infant formula. The main ingredient of cow's milk infant formula is derived from cows. "According to the Food and Agriculture Organization of the United Nations (FAO), the average global GHG (greenhouse gas) emissions from milk production, processing, and transport are estimated to be 2.4 kg $CO_2$-eq. per kg of FPCM (fat and protein corrected milk) at the farm gate" [13]. In 2007, infant formula production translated into 553 million tons of milk generating 1328 million tons of $CO_2$-eq of GHG [13]. The production of one liter of liquid cow's milk requires 800 L of water. Producing 1 kg of milk powder uses 4700 L of water and emits 21.8 kg $CO_2$-eq of GHG [13]. Based on the data, breastfeeding an infant for the first 6 months could save approximately 22.4 kg of milk powder or 105,280 L of water and 488 kg $CO_2$-eq of GHG (Estimates assume an average intake of 24–32 fluid ounces of infant formula per day and an average can of formula producing 90 fluid ounces. The second assumption is that the average can contains approximately 400 gm of powder, of which the primary ingredient in this example is dried cow's milk product. The equation used: 28 fluid ounces × 180 days/90 fluid ounces per can = 56 Cans and 56 cans × 400 gm = 22,400 gm or 22.4 kg of milk powder. Though there are other ingredients in infant formula, the bulk consists of dried cow's milk). If an average bathtub can hold 150 L of water, this equals 701 full bathtubs or 3.9 per day for the first 6 months of an infant's life. This is a crude estimation as it does not factor in the additional processing including boiling the water required to make infant formula from dried cow's milk powder. Explorative scenarios show the market for infant formula use is still expanding and can predict a directly related increase in GHG emissions.

The environmental cost of using infant formula to replace breastfeeding is more than just the carbon and water footprint of cow's milk, beginning with the deforestation of land and loss of biodiversity required for growing the cattle feed and oil palms (for palm oil). Post-production water usage is

another concern and must factor in a counter-productive process of taking liquid cows' milk, turning it into a powder and then hydrating it back into liquid. The enormous, intensive water consumption, extensive use of materials for packaging, high-demand use of energy resources in manufacturing, GHG emissions from food miles transportation, and widespread generation of household waste make infant formula production a major environmental concern and a leading contributor to global warming directly and indirectly via population growth due to increased fertility [1,13].

## 8. Conclusions

While the explicit relationship between breastfeeding and sustainability is being established, it is important to work alongside existing programs and policies. Grassroots efforts can change human behavior, attitudes, and/or intentions by increasing awareness within an enabling environment. Many projects and opportunities available today provide adults with information, health services, and resources related to breastfeeding. Examples include peer counseling programs, support groups, and parenting resources. Some schools have implemented curricula on healthy diets, exposing children of all ages to principles of sustainability, nutrition, and physical activity. These interventions provide an opportunity to reach a vast audience with important information on breastfeeding and the environment, embedding values of stewardship at every age.

Organizations such as UNICEF (United Nations Children's Fund) and the WHO (World Health Organization) lead the Global Breastfeeding Collective, a partnership exceeding 20 international agencies. As part of a call to action, governments can take seven actions to support breastfeeding and work toward a global rate of exclusive breastfeeding of at least 50% by 2025. The Global Breastfeeding Collective has also developed a country scorecard to measure indicators and targets for each of the seven actions. This is a valuable tool to inform future sustainability initiatives.

In summary, exacerbated by population growth, using infant formula to replace breastfeeding irreparably harms societies, economies, and the environment around the world. Immediate action is required from individuals, communities, industry, academia, and governments. Hence, a first and vital step to raising awareness is a need for publication of literature recognizing breastfeeding as the universal foundation of a healthy sustainable diet for women, infants, and children. Secondly, there is an urgency in assessing the global sustainability impact of using infant formula to replace breastfeeding. Finally, future endeavors should continue to demonstrate and describe the connection between breastfeeding and the triple bottom line.

**Author Contributions:** M.E.D. wrote the commentary. J.W.D. reviewed and commented on the first draft and subsequent drafts of the commentary.

**Funding:** This research received no external funding.

**Conflicts of Interest:** The authors declare no conflict of interest.

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
