# Peer review of "Breastfeeding: A Cornerstone of Healthy Sustainable Diets"

_sustainability, doi:10.3390/su11184958_

Round 1
Reviewer 1 Report
This paper is a review the role of breastfeeding in sustainable diets and the role of infant formula in unsustainable practices. It is a novel piece of work and is important to those working in this filed.
I have a few comments I would like the authors to consider.
I am not convinced there is good evidence that infant formula is the root cause of rapid population growth. You counter this argument yourself when you state (line 102-3) “mid-century population growth was due to lower death rates and unchanged birth rates. You cannot blame infant formula for the baby boom, if the birth rate did not increase.
Abstract
Line 22 – change ‘will’ to ‘could'
Line 32 – change breastfeeding to “breastmilk” – remember breast milk can be expressed, does not need to come from the breast directly.
Introduction
Line 47 – change ‘will’ to ‘could’
Line 55 – discussion of the Rio+20 is unclear, can you give some background
Line 190 – the mother would drink increased fluid for breastfeeding requirements, so the water added back to the formula would be no more than this – so this is not a real argument. But the energy needed to boil this water could be a sustainability consideration.
References
These require proof reading, there are typographical errors
Author Response
Reviewer 1:
1. First comment is addressed by removing a supposition and posing a question for the reader to consider.
"Could the evolution and permeation of formula feeding during this time frame be partly responsible for the population explosion known as the baby-boom? The Mid-Century step-increase in population growth overlapped with a cultural movement away from breastfeeding. Hence, post-War World II population momentum ran parallel to a decline in breastfeeding initiation and exclusive breastfeeding of infants under 6 months of age, compounding a vast array of public health concerns."
Abstract
Line 22 – change ‘will’ to ‘could'--done
Line 32 – change breastfeeding to “breastmilk” – remember breast milk can be expressed, does not need to come from the breast directly.--I have left breastfeeding in, as the act of breastfeeding confers health and emotional benefits separate to the breastmilk. Just feeding breastmilk in a bottle may not prevent obesity.
Introduction
Line 47 – change ‘will’ to ‘could’-done
Line 55 – discussion of the Rio+20 is unclear, can you give some background --The Rio+20 Conference on Sustainable Development Goals (SDGs)14 in 2012 developed a framework to guide implementation of sustainable development. There are seventeen SDGs that build upon the Millennium Development Goals and comprise the basis for the 2030 Agenda for Sustainable Development.
Line 190 – the mother would drink increased fluid for breastfeeding requirements, so the water added back to the formula would be no more than this – so this is not a real argument. But the energy needed to boil this water could be a sustainability consideration.--Only 500 additional calories and approximately one liter of additional drinking water per day are needed for lactation.
References
These require proof reading, there are typographical errors-corrected two.

Reviewer 2 Report
44.
While the authors are emphasising the influence of the infant formula on the population growth they should also be focusing on the western Europe situation where the exclusive breastfeeding rate is relatively low and yet the natality is negative. If the authors connect that to using other methods of contraception it is obligatory to compare advantages/disadvatages of those methods to exclusive breastfeeding.
200.
In the conclusion part the authors are emphasising the need for the urgent global food system transformation and in connection to that the need for the literature that supports breastfeeding. The problem is that new information don't necessarily follow the changes in attitudes, intentions or behaviour. The authors should be emphasising many other activities instead, that are actually already happening. Firstly, those activities are for example many projects done by UNICEF (Baby Friendly Initiative, Global Breastfeeding Collective, Mother BabyFriendly Workplace Initiative). Afterwards, it is extremely important to mention the studies that are being implemented in the obligatory health curicculum (healthy food) which is taking place in elementary, secondary schools and even colleges. The breastfeeding support projects that are happening in the community are too very relevant: Breastfeeding Support Group, Pregnancy Courses itc... Also, there are many economical and law-oriented measures that support breastfeeding (Breastfeeding at the workplace, Maternity leave, Breastfeeding in the Community, International Code of Marketing og Breast-milk Substitutes etc.)
Author Response
Reviewer 2- Thank you for your comments.
1.While the authors are emphasising the influence of the infant formula on the population growth they should also be focusing on the western Europe situation where the exclusive breastfeeding rate is relatively low and yet the natality is negative. If the authors connect that to using other methods of contraception it is obligatory to compare advantages/disadvantages of those methods to exclusive breastfeeding.
First comment is addressed by removing a supposition and posing a question for the reader to consider. I feel contraception is a separate topic for another paper.
"Could the evolution and permeation of formula feeding during this time frame be partly responsible for the population explosion known as the baby-boom? The Mid-Century step-increase in population growth overlapped with a cultural movement away from breastfeeding. Hence, post-War World II population momentum ran parallel to a decline in breastfeeding initiation and exclusive breastfeeding of infants under 6 months of age, compounding a vast array of public health concerns."
2. In the conclusion part the authors are emphasising the need for the urgent global food system transformation and in connection to that the need for the literature that supports breastfeeding. The problem is that new information don't necessarily follow the changes in attitudes, intentions or behaviour.- so true! The authors should be emphasising many other activities instead, that are actually already happening. Firstly, those activities are for example many projects done by UNICEF (Baby Friendly Initiative, Global Breastfeeding Collective, Mother BabyFriendly Workplace Initiative). Afterwards, it is extremely important to mention the studies that are being implemented in the obligatory health curicculum (healthy food) which is taking place in elementary, secondary schools and even colleges. The breastfeeding support projects that are happening in the community are too very relevant: Breastfeeding Support Group, Pregnancy Courses itc... Also, there are many economical and law-oriented measures that support breastfeeding (Breastfeeding at the workplace, Maternity leave, Breastfeeding in the Community, International Code of Marketing og Breast-milk Substitutes etc.)
Revised conclusion:
Conclusion
While the explicit relationship between breastfeeding and sustainability is being established, it is important to work alongside existing programs and policies. Grassroots efforts can change human behavior, attitudes, and/or intentions by increasing awareness within an enabling environment. Many projects and opportunities available today provide adults with information, health services, and resources related to breastfeeding. Examples include peer counseling programs, support groups, and parenting resources. Some schools have implemented curricula on healthy diets, exposing children and young adults to principles of sustainability, nutrition, and physical activity. These interventions provide an opportunity to reach a vast audience with important information on breastfeeding and the environment, embedding values of stewardship at every age.
Organizations such as UNICEF (United Nations Children’s Fund) and the WHO (World Health Organization) lead the Global Breastfeeding Collective, a partnership exceeding 20 international agencies. As part of a call to action, governments can take seven actions to support breastfeeding and work toward a global rate of exclusive breastfeeding of at least 50% by 2025. The Global Breastfeeding Collective has also developed a country scorecard to measure indicators and targets for each of the seven actions. This is a valuable tool to inform future sustainability initiatives.
In summary, exacerbated by population growth, using infant formula to replace breastfeeding irreparably harms societies, economies and the environment around the world. Immediate action is required from individuals, communities, industry, academia, and governments. Hence, a first and vital step to raising awareness is a need for publication of literature recognizing breastfeeding as the universal foundation of a healthy sustainable diet for women, infants, and children. Secondly, there is an urgency in assessing the global sustainability impact of using infant formula to replace breastfeeding. Finally, future endeavors should continue to demonstrate and describe the connection between breastfeeding and the triple bottom line.
